# Optimal Solution for Frequency and Voltage Control of an Islanded Microgrid Using Square Root Gray Wolf Optimization

Aftab Ahmed Almani [1,*], XueShan Han [1], Farhana Umer [2], Rizwan ul Hassan [3], Aamir Nawaz [4], Aamer Abbas Shah [5] and Ehtasham Mustafa [4]

1    Key Laboratory of Power System Intelligent Dispatch and Control, Shandong University, Jingshi Road #17923, Jinan 250000, China
2    Department of Electrical Engineering, The Islamia University of Bahawalpur, Bahawalpur 63100, Pakistan
3    NFC Institute of Engineering and Fertilizer Research, Faisalabad 38000, Pakistan
4    Faculty of Engineering and Technology, Gomal University, Dera Ismail Khan 29050, Pakistan
5    School of Electrical and Control Engineering, Xuzhou University of Technology, Xuzhou 221018, China
*    Correspondence: aaftabalmani@yahoo.com

**Abstract:** Voltage and frequency deviation in the islanded operation of a microgrid (MG), due to the uncertainty and lack of inertia in the selection of optimal proportional integral (PI) controller gain, is a challenging task. Although various optimization algorithms have been proposed to achieve this task, most of them require a large number of iterations and are time intensive, making them inefficient for real-time applications. Gray wolf optimization (GWO), a new meta-heuristic algorithm, addresses these issues and has many advantages, including simplicity due to fewer control parameters, flexibility, and globalism. This paper proposes a simple and efficient modified algorithm, called square root gray wolf optimization (SRGWO) algorithm, to realize superior hunting performance. SRGWO is verified using twenty-three benchmark test functions. The algorithm is applied for optimal voltage and frequency regulation of a photovoltaic-based microgrid system operating in the islanded mode during distributed generation insertion and load change conditions. The voltage and frequency gain parameters of the PI controller are optimized. A comparison of the simulation results of the SRGWO algorithm with those of the original gray wolf algorithm (GWO), particle swarm optimization (PSO), augmented gray wolf optimization (AGWO), enhanced gray wolf optimization (EGWO), and gravitational search algorithm (GSA) reveal that the proposed SRGWO algorithm significantly improves system performance while maintaining its simplicity and easy implementation. Furthermore, the SRGWO algorithm obtains the minimum fitness function value in fewer iterations than other algorithms. Moreover, it improves the power quality of the system with regard to minimum total harmonic distortion.

**Keywords:** microgrid; gray wolf optimization; optimal voltage and frequency control; renewable energy sources

## 1. Introduction

Nowadays, power systems depend on utility. Heavy power loss often occurs at peak hours. In order to avoid such power loss, ensuring the stability of power generation stations is crucial [1,2]. The demand side management of smart grids allows the customer to control their power consumption depending on their load conditions and services [3]. Microgrids (MGs) are constantly connected to the primary electrical grid under normal conditions, and when a shortage of power occurs, they disconnect to an isolate mode. One of the most important features and advantages of using a microgrid is its ability to be islanded and operate independently. The advantage of island mode lies in its abillity to improve the reliability and power quality of real customers in the MG [4]. It consists of renewable and

non-renewable energy resources. The loads in the distribution system, are able to operate both as grid-connected and islanded modes [5]. The voltage and frequency of the system are controlled by the power grid in the grid-connected mode operation. Distributed generators (DGs) inject almost fixed power to the grid [6], which contributes to the economic and secure operation of the power system [6,7]. Generally, voltage and frequency instability is attributed to the large deviation between generated electrical energy and utilized power [8]. Some control strategies with efficient algorithms have been developed for the stability of voltage and frequency to ensure the durability of services [9]. In both modes, the acceptable levels of main parameters can be achieved with respect to the power excellence of the MG [10]. In a conventional power system, the voltage and frequency control can be achieved easily because mostly the power consumption side produces the uncertainties. By contrast, in MGs, due to the difference in power generation and load, functional complexity, structural variation, and changing the nature of renewable energy sources (RES), controlling the voltage and frequency is a complex task [11,12]. Conventional regulators cannot deliver acceptable achievements in some serious functioning situations given the rapidly changing operating conditions of MG. Therefore, voltage and frequency control require a powerful and intelligent controller in an islanded MG [13].

Because the voltage and frequency variations of MG occur mostly in the islanded mode rather than the grid-connected mode, an effective control scheme should be able to sustain the reliability and quality of power supply to the load throughout. The main objective is to preserve, within the acceptable limit, the voltage and frequency of the MG in the islanded mode. There are two power control strategies in the DGs: the voltage–frequency control strategy in the islanded mode, or the active–reactive power control strategy in the grid-connected mode. Several control schemes using different algorithms have been proposed to regulate the frequency and voltage of the MG [14]. But the frequency and voltage of the MG in islanded vary and need autonomous control [5]. In [15], the control loop of power was achieved by the control of the first loop. In [16], a control loop of voltage was used for controlling the main parameters of the MG islanded mode. The reference voltage and reference frequency were received from the voltage controller to the current control loop by reducing the voltage variation using proportional integral (PI) controllers. However, a major drawback of PI controllers is their partial execution because they depend on the precise tuning of their gain coefficients ($K_p$ and $K_i$), which are either established as static during the procedure or calculated to obtain dynamic values. When the static PI gains controller is employed for a voltage control loop, the adaptive or "trial and error" technique [9,17,18] and the Ziegler–Nichols technique [19–21] can be used to determine the gain parameters. However, the use of these methods adds to the total time spent in control activities, which might result in considerable delays in unstable operating areas. Hence, the suitable tuning of PI gains is extremely important to improve system efficiency. Nowadays, artificial intelligence (AI) has been adapted into an advanced method to improve the changeable response of MG systems [22,23]. AI-based controllers employed on MG systems confirm better combination and separation of DGs in present electrical grid systems, improve the voltage and frequency of transient response throughout changing the loads, enhance end-user electrical energy control, and ensure transient stability of the MG system. Furthermore, intelligent research methods obtaining optimal solutions yield better problem-solving results than traditional mathematical techniques [24]. Various AI-based techniques have been developed to control the voltage and frequency of MG systems. Further, these techniques did not utilize time-intensive and insufficient conventional PI tuning techniques; for example, fuzzy logic (FL) [25], genetic algorithms (GAs), [24,26], particle swarm optimization (PSO) [13,24], and gravitational search algorithms (GSAs) [14,27,28].

However, AI-based algorithms such as GA, PSO, and FL have a few disadvantages. For instance, GA solves the local convergence rather than solving the global, which requires the translation of groups of dynamic data in the modern MG controls into conventional optimization techniques [15]. PSO also results in a small amount of convergence in the reiterative procedure, limited by a local minimum in a large space [29] and the ambiguity

in the choice of its parameters [30]. Although it works relatively well during the initial iterations, optimal results cannot be obtained within limited reference functions [31,32]. Great improvement in heuristic algorithms has inspired researchers to apply them in different fields of power system optimization. Grey wolf optimization (GWO) is the latest successful meta-heuristic algorithm that emulates the gray wolf societal pecking order demeanor. A pack of gray wolves, of an average pack size of five to twelve, are categorized as top predators, who fully trust the leadership chain and hunting demeanors [33]. Enhanced gray wolf optimization (EGWO) uses parameter $\alpha$, a random number having a value within 0–1, and an alpha search agent. However, the alpha is not suitable for the higher order of the gray wolves' performance and will lead to inactivity in certain local solutions [34]. Modified GWO (mGWO) is a balanced exchange of the exploration and exploitation in original GWO algorithm. However, its suitability for power systems is questionable [35]. Augmented gray wolf optimization (AGWO) is an improved GWO algorithm with regard to exploration and exploitation. To avoid stagnation, exploration is enhanced by making nonlinear reducing parameters from 2–1. The exploitation is expanded by updating search agent rankings with the average positions of the alpha (first best ranking) and the betas (second best ranking) [36]. However, this algorithm is very slow and can be implemented only on the grid-connected mode.

This study proposes a novel modified GWO algorithm called square root gray wolf optimization (SRGWO), which shows improved performance without compromising its interpretability and robustness. The SRGWO algorithm is an efficient meta-heuristic algorithm that emulates the gray wolf societal hierarchy demeanor. Firstly, the proposed algorithm was verified with twenty-three benchmark functions and achieved better results than other algorithms. The proposed algorithm solves the problem of determining optimal parameters of PI controllers to control the voltage and frequency of the MG in an islanded mode under the conditions of inserting the DG and changing the load. To confirm the efficiency of this method, its performance for voltage–frequency regulation was compared with the performance of the regulators of GWO, AGWO, and EGWO under similar working conditions.

Further, the combination of SRGWO and the PI controller was used to design the islanded MG system using the voltage and frequency regulate technique. To the best of our knowledge, this is the first time the tuning of the PI controller with SRGWO has been attempted. To achieve the basic advantage of both controllers, SRGWO was used to tune the control parameters of the PI controller in this work. Herein, as a replacement for gain arrangement, the gains of the controller were tuned according to the operating situations. The effective strategy of using a PI controller for advance power systems meets the following stipulations:

1.  Efficacious against load disturbances
2.  Resilient against RES uncertainties
3.  Least sensitive to MG uncertainties
4.  Adjustable and flexible in tuning the parameters according to system operating situations.

To ensure that the proposed system meets these specifications, the performance of the proposed controller for the MG system under various operating scenarios was analyzed. Furthermore, the accomplishment of the proposed controller was compared to the numerous powerful methods in the literature.

The main contributions of this paper are as follows:

1.  GWO was modified and analyzed with respect to the standard benchmark functions, comparing with the main GWO algorithm and additional modified GWO, PSO, and GSA algorithms.
2.  The parameters of PI controller were optimized using the proposed SRGWO algorithm.
3.  The PI controller parameters tuned with SRGWO were implemented for the voltage and frequency control of an islanded MG system, and its performance was compared with original GWO, AGWO, and EGWO.

4.  The robustness of the proposed controller against RES uncertainties and MG uncertainties with diverse operating scenarios in a single framework was tested.

The paper is ordered as follows. The original GWO is described in Section 2. The novel GWO algorithm and tested benchmark functions are described in Section 3. The MG and mathematical modeling of the three-phase islanded technique, along with its control architecture, are explained in Section 4. In Section 5, the problem formulation is described. In Section 6, the results and discussion are presented. Finally, conclusions are presented in Section 7.

## 2. Grey Wolf Optimization Algorithm

A modern meta-heuristic algorithm is known as the GWO algorithm, and mimics behavior of the gray wolves, which live in packs with an average of five to twelve members. In the gray wolf pack, a rigorous pecking order is exercised, wherein the pack has a captain named the alpha ($\alpha$), accompanied by inferior wolves named betas ($\beta$) that help the alpha ($\alpha$) in decisions, along with $\delta$ and $\omega$, as illustrated in Figure 1 [33,37].

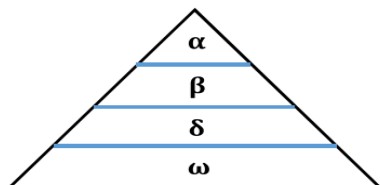

**Figure 1.** Grey wolf pecking order.

The method of chasing a quarry using grey wolves is conducted in four parts: probing for the quarry, encompassing the quarry, then chasing and attacking it. The exact model of encompassing quarry is articulated as follows:

$$\vec{D} = \left| \vec{C}.\vec{X}_{pi} - \vec{X}_i \right| \tag{1}$$

$$\vec{X}_{t+1} = \vec{X}_{pi} - \vec{A}.\vec{D} \tag{2}$$

where the position vector of the gray wolf is denoted by $\vec{X}_i$, the position vector of the quarry is denoted by $\vec{X}_{pi}$ and t represents the current iteration. The $\vec{A}$ coefficient vector and the $\vec{C}$ coefficient vector are calculated as follows:

$$\vec{\alpha} = 2 - 2 * \frac{t}{Max.iter} \tag{3}$$

$$\vec{A} = 2\,\vec{a}.\vec{r_1} - \vec{a} \tag{4}$$

$$\vec{C} = 2.\vec{r_2} \tag{5}$$

where, $r_1$ and $r_2$ are uniformly and randomly distributed vectors ranging from 0–1. The convergence factor $\vec{\alpha}$ represents a value that gradually reduces from 2-0 through iteration (*t*), and iterates until it reaches maximum iteration (Max: iter). The probing of the quarry position could be attained using the discrepancy of exploration agents, which could be attained while $|A| > 1$. The exploitation of the quarry could be obtained from the converging exploration agents that is studied when $|A| < 1$. The hunting is controlled through agent $\alpha$, and with complete assistance from agents $\beta$ and $\delta$, as follows:

$$\vec{D_\alpha} = \left| \vec{C_1}.\vec{X}_{\alpha i} - \vec{X}_i \right|, \ \vec{D_\beta} = \left| \vec{C_2}.\vec{X}_{\beta i} - \vec{X}_i \right|, \ \vec{D_\delta} = \left| \vec{C_1}.\vec{X}_{\delta i} - \vec{X}_i \right| \tag{6}$$

$$\vec{X_1} = \vec{X_{\alpha i}} - \vec{A_1}.\vec{D_\alpha},\ \vec{X_2} = \vec{X_{\beta i}} - \vec{A_2}.\vec{D_\beta}\ ,\ \vec{X_3} = \vec{X_{\delta i}} - \vec{A_1}.\vec{D_\delta} \tag{7}$$

$$\vec{X_{t+1}} = \frac{\vec{X_1} + \vec{X_2} + \vec{X_3}}{3} \tag{8}$$

where, $\vec{X_{\alpha i}}$ represents the position vector of the α wolf, $\vec{X_{\beta i}}$ represents the position vector of the β wolf, and $\vec{X_{\delta i}}$ is the position vector of the δ wolf. $\vec{X_{t+1}}$ represents the updated position of the gray wolves.

### 3. Square Root Grey Wolf Optimization Algorithm

The massive power system applications, such as islanded-mode PV solar plants, are nonlinear models, and it is therefore hard to catch the transfer function for optimum performance. Thus, the online optimization of electrical power systems is the best other solution in place of the transfer function. Thus, similar to many proposed algorithms such as PSO, the GWO algorithm can be enhanced and adapted for better performance in exploration and exploitation in various technical field applications. Moreover, concerning the optimization methods based on population, the most appropriate way to converge towards the global minimum may have a two-phase division, fundamentally. During the initial optimization stages, the individuals can be stimulated to scatter through the entire search space.

During the later phase, the individuals should utilize information collected to converge on the global minimum. The current study proposes a novel modification in order to enhance the probing capability of the GWO algorithm, which is called square root gray wolf optimization (SRGWO). With the help of fine adjustment of the parameters $\vec{a}$ and $\vec{A}$, the two phases can be balanced to discover the global minima with fast convergence. Though various adjustments of individual-based algorithms encourage local optima avoidance, in the proposed algorithm, the adaptive values of $\vec{a}$ and $\vec{A}$ generate the transition between exploration and exploitation, where half of the iterations are attached to exploration $(|A| > 1)$ while the rest are meant for exploitation $(|A| < 1)$.

Normally, higher exploration of search space produces lower probabilities in order to improve the exploration rate in which exponential functions are employed in place of linear function for decreasing the value of $\vec{a}$ over the course of iterations. It is likely that over-exploration may produce excessive randomness and fail to submit ideal optimization results. Similarly, excessive exploitation has been found to produce insufficient randomness and fails to yield effective optimization results. Hence, for the sake of ideal optimization results, a balance between exploration and exploitation appears to be fundamental. The proposed algorithm uses the function of square root for the decay of $\vec{a}$ over the course of iterations. In SRGWO, all parameters must measure for exploration and exploitation in parameter $\vec{A}$, which mostly relies on parameter $\vec{a}$, as is the case in (4). In the proposed algorithm, the parameter varies nonlinearly and randomly from 2-1 as in (9). Hence, the possibilities of exploration state and exploitation state are balanced.

$$\vec{a} = 2 - 2 * \left( \frac{l^{\frac{1}{2}}}{Max.iter^{\frac{1}{2}}} \right) \tag{9}$$

$$\vec{A} = 2\vec{a}.\vec{r_1} - \vec{a} \tag{10}$$

$$\vec{C} = 2.\vec{r_2} \tag{11}$$

Hunting and decision-making rely on the modernizing of alpha ($\alpha$), beta ($\beta$) and delta ($\delta$) in GWO algorithms as in 6-8. However, the hunting will depend only on the two top members $\alpha$ and $\beta$ in the proposed SRGWO algorithm, as follows:

$$\vec{D}_\alpha = \left| \vec{C}_1.\vec{X}_{\alpha i} - \vec{X}_i \right|, \ \vec{D}_\beta = \left| \vec{C}_2.\vec{X}_{\beta i} - \vec{X}_i \right| \tag{12}$$

$$\vec{X}_1 = \vec{X}_{\alpha i} - \vec{A}_1.\vec{D}_\alpha, \ \vec{X}_2 = \vec{X}_{\beta i} - \vec{A}_2.\vec{D}_\beta \tag{13}$$

$$\vec{X}_{t+1} = \frac{\vec{X}_1 + \vec{X}_2}{2} \tag{14}$$

*Benchmark Functions*

The 23 standard benchmark functions were tested using the proposed SRGWO algorithm. In [33], all 23 benchmark functions were given in tabular form. The GWO, PSO, AGWO, EGWO, and GSA algorithms were compared with the proposed algorithm for all benchmark functions. In every simulation, the search agents and maximum iterations were 30 and 500, respectively, and the executions number is thirty times to calculate average and standard deviation of all algorithms, which are given in Tables 1–3. Simulation results are shown in Table 1, where the proposed SRGWO algorithm achieved the five best results of seven. Hence, the obtained results verified the exploitability of the proposed algorithm over other nominated algorithms. The obtained results presented in the other two tables illustrate that the proposed algorithm is enhanced in exploration mode compared with the five other algorithms. The convergence characteristics of the SRGWO algorithm was compared with other algorithms, as illustrated in Figure 2. Furthermore, the SRGWO algorithm achieved the best results for 12 out of 23 benchmark functions, while the PSO and GSA algorithms achieved the best results for 4 out of 23, and the GWO algorithm achieved the best results for 3 out of 23. Lastly, the AGWO and EGWO algorithms achieved the best result for 1 out of 23 functions, as shown in Tables 1–3. The high-dimensional nature of the benchmark functions were mostly verified, and the proposed algorithm obtained results that were better than the other algorithms, as shown in Table 1. Further, Tables 1 and 2 shows unimodal and multimodal functions. The gray color boxes indicate the best results. Moreover, Table 3 shows the evolution of functions F14-F23 for fixed dimensions of multimodal functions.

**Table 1.** Simulation results of uni-modal functions.

| F | GWO | | PSO | | AGWO | | EGWO | | GSA | | SRGWO | |
|---|---|---|---|---|---|---|---|---|---|---|---|---|
| | Ave | Std | Ave | Std | Ave | Std | Ave | Std | Ave | Std | Ave | Std |
| F1 | $1.7 \times 10^{-24}$ | $3.98 \times 10^{-24}$ | $1.56 \times 10^{-4}$ | $1.45 \times 10^{-4}$ | $1.53 \times 10^{-43}$ | $2.86 \times 10^{-43}$ | $1.57 \times 10^{-30}$ | $1.58 \times 10^{-29}$ | $1.95 \times 10^{-16}$ | $8.42 \times 10^{-17}$ | $1.27 \times 10^{-197}$ | $0$ |
| F2 | $2.25 \times 10^{-15}$ | $4.42158 \times 10^{-17}$ | $7.89 \times 10^{-2}$ | $1.49 \times 10^{-1}$ | $4.8 \times 10^{-27}$ | $5.51 \times 10^{-27}$ | $8.05 \times 10^{-20}$ | $7.28 \times 10^{-20}$ | $2.15 \times 10^{-6}$ | $9.24 \times 10^{-06}$ | $1.27 \times 10^{-119}$ | $5.04 \times 10^{-119}$ |
| F3 | $1.19 \times 10^{-5}$ | $2.85 \times 10^{-5}$ | $7.7 \times 10^{1}$ | $3.05 \times 10^{1}$ | $9.91 \times 10^{-8}$ | $4.33 \times 10^{-7}$ | $1.14 \times 10^{-4}$ | $3.154 \times 10^{-4}$ | $9.49 \times 10^{2}$ | $4.05 \times 10^{2}$ | $2.15 \times 10^{-123}$ | $6.77 \times 10^{-123}$ |
| F4 | $7.77 \times 10^{-7}$ | $1.29 \times 10^{-6}$ | $1.13$ | $2.41 \times 10^{-1}$ | $1.11 \times 10^{-11}$ | $2.21 \times ^{-10}$ | $2.33 \times 10^{-1}$ | $9.12 \times 10^{-1}$ | $7.4$ | $2.6878$ | $3.42 \times 10^{-74}$ | $1.53 \times 10^{-73}$ |
| F5 | $2.74 \times 10^{1}$ | $5.62 \times 10^{-1}$ | $7.88 \times 10^{1}$ | $7.75 \times 10^{1}$ | $2.68 \times 10^{1}$ | $4.55 \times 10^{1}$ | $2.80 \times 10^{1}$ | $8.82 \times 10^{-1}$ | $5.63 \times 10^{1}$ | $3.93 \times 10^{1}$ | $2.84 \times 10^{1}$ | $5.78 \times 10^{-1}$ |
| F6 | $1.5874$ | $3.38 \times 10^{-1}$ | $1.5 \times 10^{-4}$ | $1.59 \times 10^{-4}$ | $1.4759$ | $4.24 \times 10^{-1}$ | $3.6435$ | $3.86 \times 10^{-1}$ | $8.65$ | $1.03 \times 10^{1}$ | $5.6343$ | $0.1677 \times 10^{-1}$ |
| F7 | $1.66 \times 10^{-3}$ | $1.20 \times 10^{-3}$ | $2.86 \times 10^{-1}$ | $3.06 \times 10^{-1}$ | $1.60 \times 10^{-3}$ | $1.24 \times 10^{-3}$ | $7.80 \times 10^{-3}$ | $6.19 \times 10^{-3}$ | $8.61 \times 10^{-2}$ | $3.47 \times 10^{-2}$ | $1.04 \times 10^{-4}$ | $9.11 \times 10^{-5}$ |

**Table 2.** Simulation results of multi-modal functions.

| F | GWO | | PSO | | AGWO | | EGWO | | GSA | | SRGWO | |
|---|---|---|---|---|---|---|---|---|---|---|---|---|
| | Ave | Std | Ave | Std | Ave | Std | Ave | Std | Ave | Std | Ave | Std |
| F8 | $-5.50 \times 10^{3}$ | $9.71 \times 10^{2}$ | $-4.84 \times 10^{3}$ | $1.14 \times 10^{3}$ | $-3.78 \times 10^{3}$ | $3.30 \times 10^{2}$ | $-6.518 \times 10^{3}$ | $6.19 \times 10^{2}$ | $-2.520 \times 10^{3}$ | $3.95 \times 10^{2}$ | $-2.568 \times 10^{3}$ | $3.07 \times 10^{2}$ |
| F9 | $1.3$ | $2.16$ | $5.53 \times 10^{1}$ | $1.48 \times 10^{1}$ | $0$ | $0$ | $1.68 \times 10^{2}$ | $5.118 \times 10^{1}$ | $3.14 \times 10^{1}$ | $6.63$ | $0$ | $0$ |
| F10 | $2.26 \times 10^{-13}$ | $3.03 \times 10^{-13}$ | $2.90 \times 10^{-1}$ | $5.40 \times 10^{-1}$ | $8.88 \times 10^{-15}$ | $2.27 \times 10^{-15}$ | $4.50 \times 10^{-1}$ | $1.11$ | $4.66 \times 10^{-2}$ | $1.02$ | $8.88 \times 10^{-16}$ | $2.02 \times 10^{-31}$ |
| F11 | $5.05 \times 10^{-3}$ | $9.52 \times 10^{-3}$ | $6.55 \times 10^{-3}$ | $7.46 \times 10^{-3}$ | $4.89 \times 10^{-4}$ | $2.19 \times 10^{-3}$ | $1.48 \times 10^{-2}$ | $2.12 \times 10^{-2}$ | $2.53 \times 10^{1}$ | $5.93$ | $0$ | $0$ |
| F12 | $4.94 \times 10^{-3}$ | $1.59 \times 10^{-2}$ | $1.10 \times 10^{-2}$ | $3.18 \times 10^{-2}$ | $1.70 \times 10^{-1}$ | $3.7 \times 10^{-1}$ | $1.99$ | $2.4$ | $1.97$ | $1.21$ | $7.00 \times 10^{-1}$ | $1.10 \times 10^{-1}$ |
| F13 | $1.31$ | $1.90 \times 10^{-1}$ | $4.46 \times 10^{-3}$ | $5.52 \times 10^{-3}$ | $1.13$ | $2.30 \times 10^{-1}$ | $2.5$ | $4.30 \times 10^{-1}$ | $7.36$ | $6.56$ | $2.6$ | $7.00 \times 10^{-2}$ |

**Table 3.** Simulation results of fixed-dimension multi-modal functions.

| F | GWO | | PSO | | AGWO | | EGWO | | GSA | | SRGWO | |
|---|---|---|---|---|---|---|---|---|---|---|---|---|
| | Ave | Std | Ave | Std | Ave | Std | Ave | Std | Ave | Std | Ave | Std |
| F14 | $11.16$ | $2.07$ | $11.46$ | $8.445$ | $7.15$ | $4.14$ | $14.14$ | $3.44$ | $6.39$ | $2.79$ | $2.98$ | $4.8 \times 10^{-16}$ |
| F15 | $1.2 \times 10^{-2}$ | $1.04 \times 10^{-2}$ | $1.7 \times 10^{-3}$ | $1.32 \times 10^{-3}$ | $9.18 \times 10^{-3}$ | $1.06 \times 10^{-2}$ | $3.59 \times 10^{-3}$ | $7.67 \times 10^{-3}$ | $8.26 \times 10^{-3}$ | $7.56 \times 10^{-3}$ | $6.81 \times 10^{-4}$ | $1.16 \times 10^{-4}$ |
| F16 | $-1.03$ | $4.56 \times 10^{-16}$ | $-1.03$ | $4.56 \times 10^{-16}$ | $-1.03$ | $4.56 \times 10^{-16}$ | $-1.03$ | $4.56 \times 10^{-16}$ | $-1.03$ | $4.56 \times 10^{-16}$ | $-1.03$ | $4.56 \times 10^{-16}$ |
| F17 | $0.3978$ | $4.10 \times 10^{-6}$ | $0.3978$ | $2.98 \times 10^{-5}$ | $0.39789$ | $2.68 \times 10^{-5}$ | $0.39827$ | $6.13 \times 10^{-4}$ | $3.98 \times 10^{-1}$ | $0$ | $3.99 \times 10^{-1}$ | $9.73 \times 10^{-4}$ |
| F18 | $3.0001$ | $3.08 \times 10^{-5}$ | $3$ | $0$ | $3.00005$ | $2.24 \times 10^{-5}$ | $7.05$ | $18.1121$ | $3$ | $0$ | $3$ | $0$ |
| F19 | $-3.8624$ | $1.28 \times 10^{-3}$ | $-3.8628$ | $4.47 \times 10^{-5}$ | $-3.86032$ | $2.16 \times 10^{-3}$ | $-3.8627$ | $6.16 \times 10^{-5}$ | $-3.8628$ | $1.37 \times 10^{-15}$ | $-3.85404$ | $3.33 \times 10^{-3}$ |
| F20 | $-3.2841$ | $7.99 \times 10^{-2}$ | $-3.2802$ | $5.84 \times 10^{-2}$ | $-3.1152$ | $0.221416$ | $-3.2707$ | $8.05 \times 10^{-2}$ | $-3.322$ | $1.37 \times 10^{-15}$ | $-2.7921$ | $0.3957$ |
| F21 | $-9.64$ | $1.56$ | $-6.63$ | $3.39$ | $-6.25$ | $1.91$ | $-6.77$ | $3.56$ | $-7.26$ | $3.65$ | $-3.81$ | $1.52$ |
| F22 | $-8.98$ | $4.87$ | $-8.51$ | $3.02$ | $-7.42$ | $1.77$ | $-7.26$ | $3.63$ | $-10.17$ | $1.06$ | $-4.07$ | $1.38$ |
| F23 | $-9.72$ | $1.98$ | $-5.76$ | $6.49$ | $-7.04$ | $2.1$ | $-8.1$ | $3.83$ | $-10.54$ | $8.47 \times 10^{-15}$ | $-4.42$ | $0.85$ |

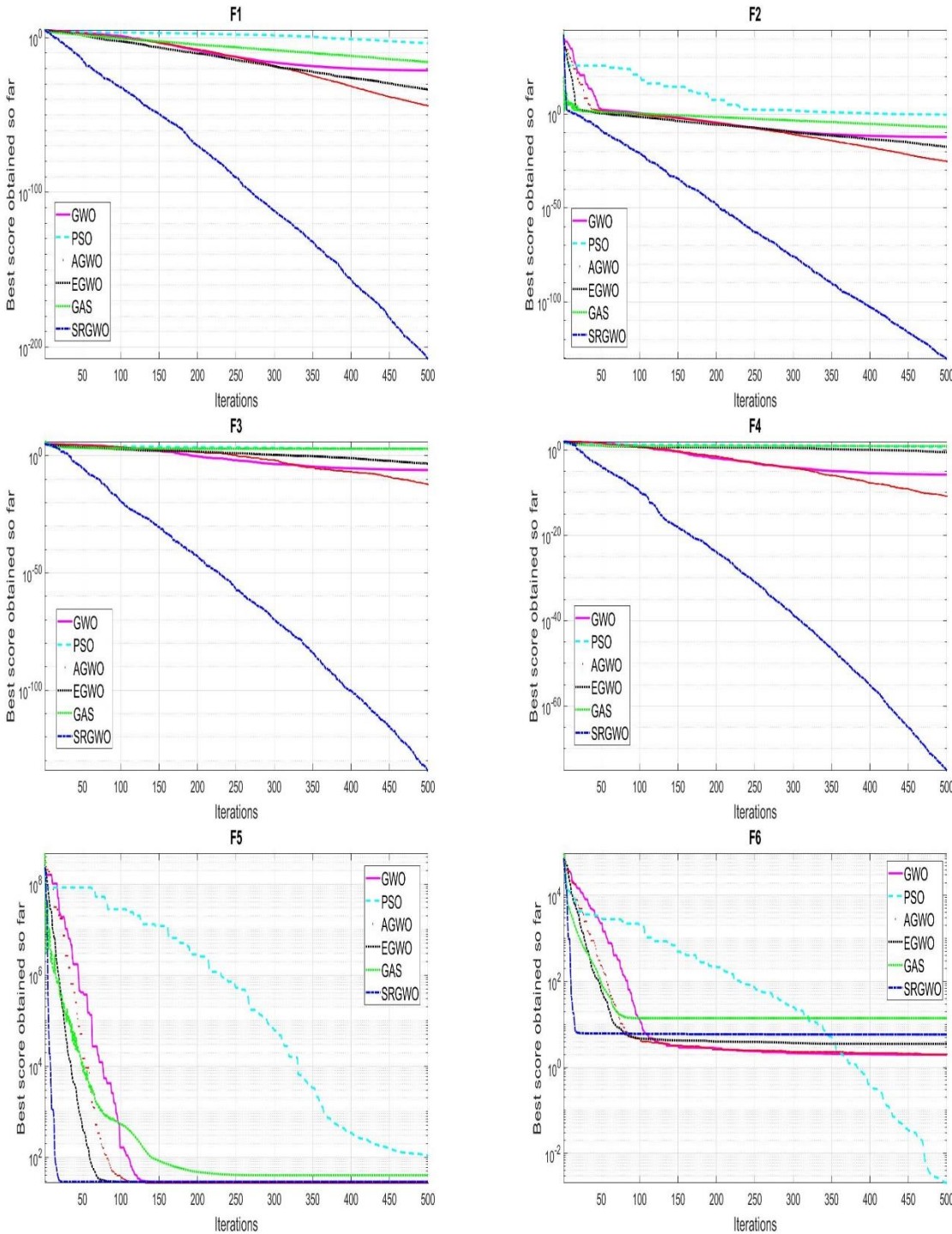

**Figure 2.** *Cont.*

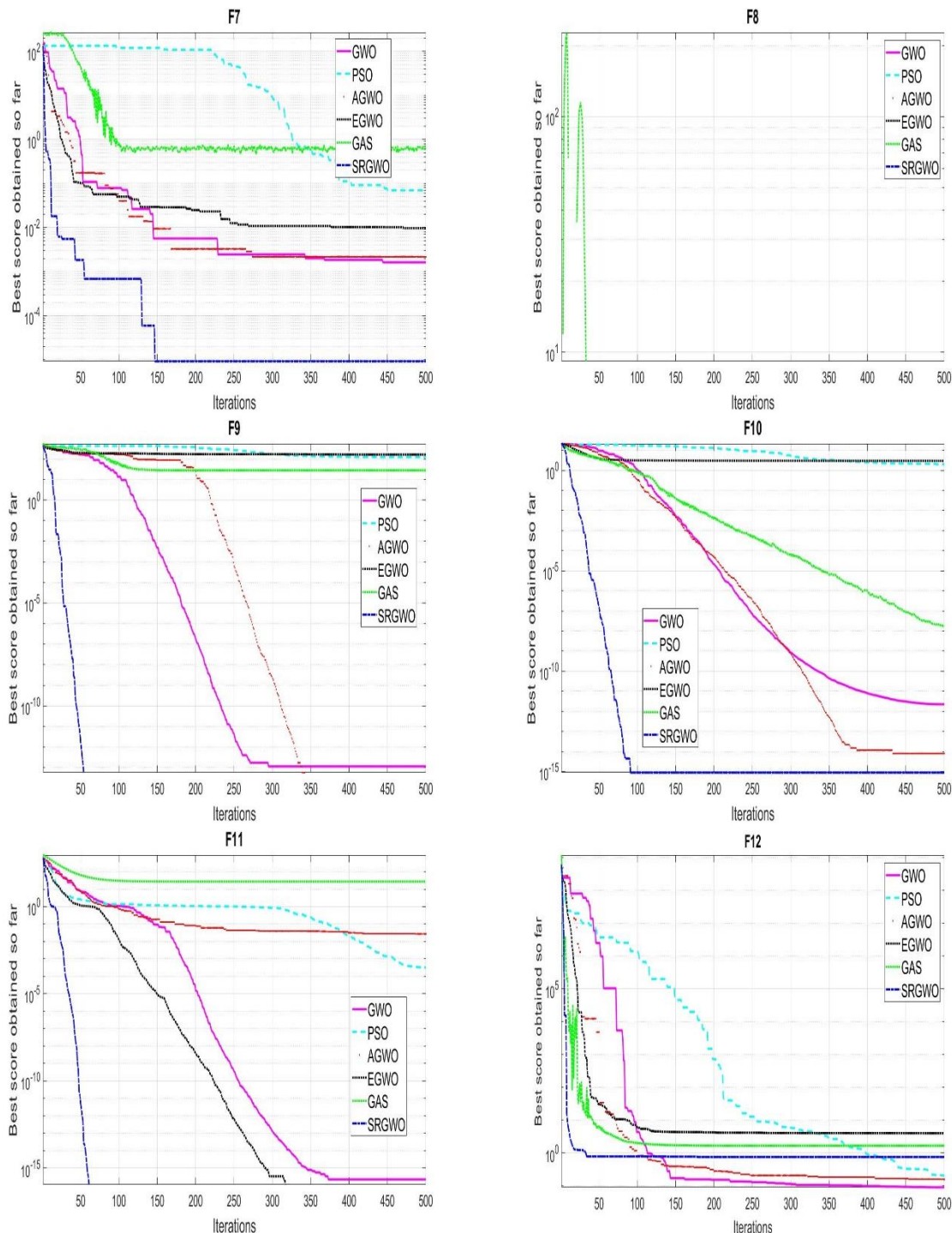

**Figure 2.** *Cont.*

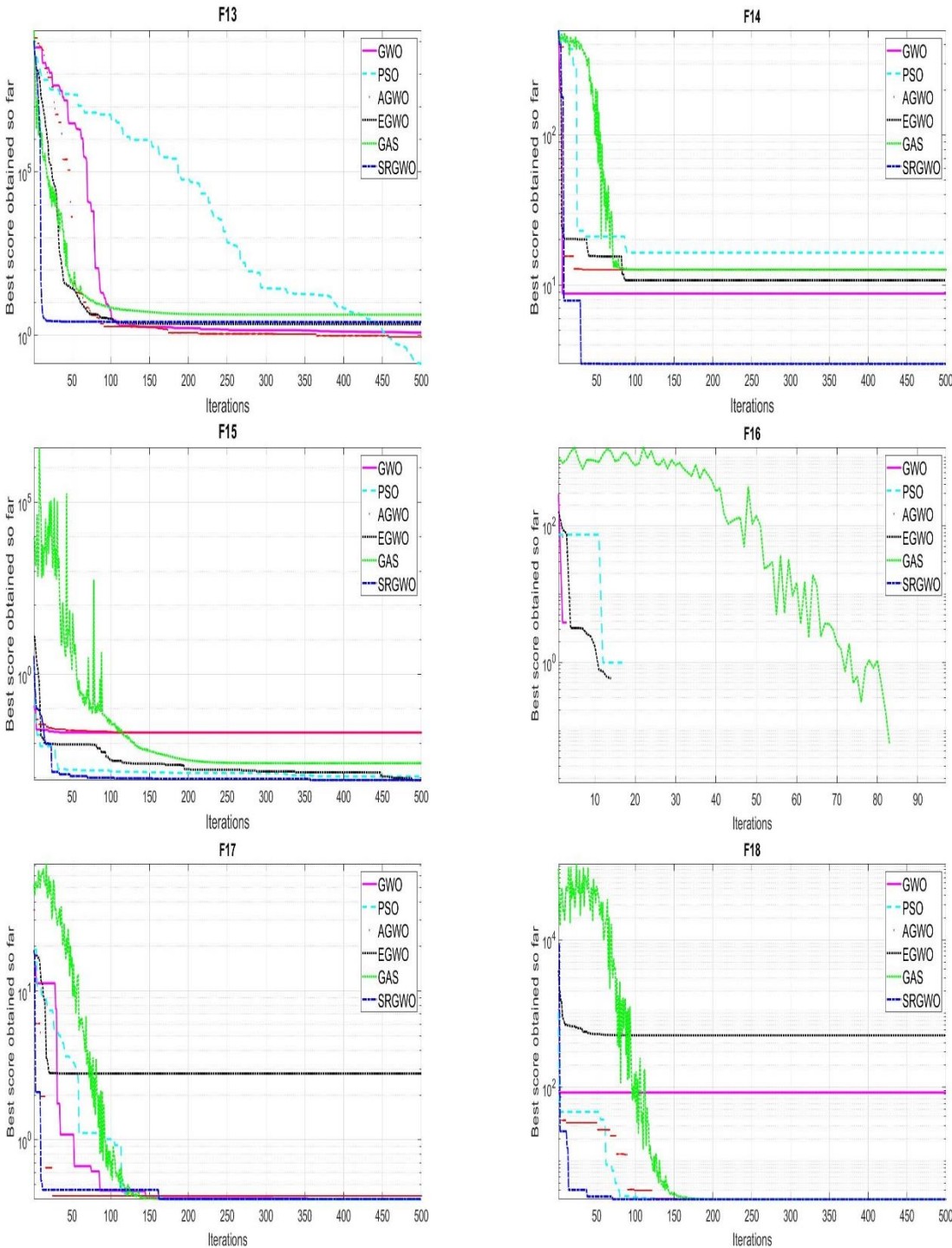

**Figure 2.** *Cont.*

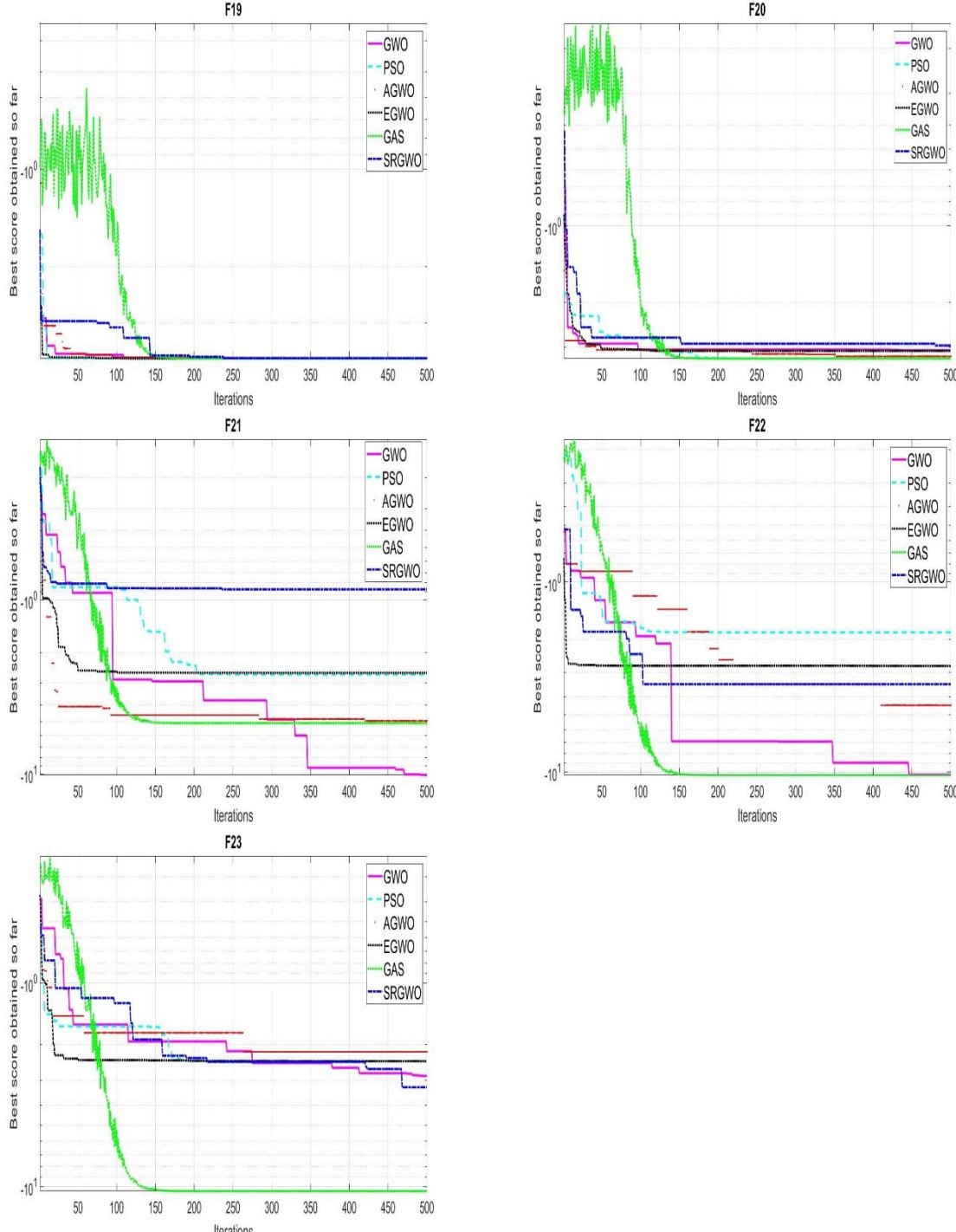

**Figure 2.** Convergence curves of the GWO, PSO, AGWO, EGWO, GSA, and SRGWO algorithms for twenty-three benchmark functions.

A statistical test based on the non-parametric sign test [37] was used to evaluate the proposed SRGWO algorithm against GWO, PSO, AGWO, EGWO, and GSA. The non-parametric symbols "-", "+", and "≈" demonstrate whether the presentation of SRGWO is statistically inferior to, superior to, or similar to the second optimizer, respectively, in Table 4. Hence, as Table 4 shows, the results' difference verified the superiority of the proposed algorithm compared with the others. The total non-parametric sign test result

was 115. The number of results for "+" was 60/115, for "-" was 46/115, and for "≈" was 9/115.

**Table 4.** Non-parametric sign test results.

| F | SRGWO vs. GWO | SRGWO vs. PSO | SRGWO vs. AGWO | SRGWO vs. EGWO | SRGWO vs. GSA |
|---|---|---|---|---|---|
| F1 | + | + | + | + | + |
| F2 | + | + | + | + | + |
| F3 | + | + | + | + | + |
| F4 | + | + | + | + | + |
| F5 | - | + | - | - | - |
| F6 | - | - | - | - | - |
| F7 | + | + | + | + | + |
| F8 | - | + | - | - | - |
| F9 | + | - | ≈ | ≈ | + |
| F10 | + | + | + | + | + |
| F11 | + | - | + | + | + |
| F12 | - | - | - | - | + |
| F13 | - | - | - | - | - |
| F14 | + | + | + | + | + |
| F15 | + | + | + | + | + |
| F16 | ≈ | ≈ | + | + | ≈ |
| F17 | - | - | ≈ | ≈ | - |
| F18 | + | ≈ | + | + | + |
| F19 | + | - | - | - | - |
| F20 | + | - | - | - | - |
| F21 | + | - | - | - | - |
| F22 | + | - | - | - | - |
| F23 | + | - | - | - | - |
| No. of + | 16 | 10 | 11 | 11 | 12 |
| No. of − | −6 | −11 | −10 | −9 | −10 |
| No of ≈ | 1 | 2 | 2 | 2 | 2 |

## 4. Microgrid Modelling

An MG is essentially a group of loads that are delivered from small sources of energy, e.g., wind turbines, microturbines, solar photovoltaics, and fuel cells, working as the only controlled system that is able to deliver heat and energy to the definite part [38]. In addition, for the grid-connected mode with MG running the main grid, the voltage and frequency of the system is regulated using a huge electrical system. In addition to keeping the load balanced, regulating frequency and voltage is very important for MG operation in islanded mode [39]. This is due to the fact that in MG systems with huge infiltrations of DGs, high voltage and frequency fluctuations can occur throughout the insertion of DGs or load changes. Furthermore, the control system of MG confirms, as a prerequisite, the absence of huge circulating reactive currents from small energy sources. The minor errors at fixed points of voltage and frequency due to circulating currents exceed the level of small energy sources [40]. High-switching frequency pulses and problems of power quality are issues owing to the application of power electronic converters [41]. Therefore, MG faces serious problems related to power quality, especially when assimilating a large number of DGs [42,43]. It is relevant to indicate the nonexistence of instability and ambiguity in the sample of optimal PI parameters. Further, these parameters are utilized to gain control amplifications due to huge deviations in power, voltage, and frequency level in the islanded mode. Hence, in this research, we discuss these issues as well as improve MG's achievements in the islanded operating approach.

This section may be divided by subheadings. It should provide a concise and precise description of the experimental results, their interpretation, as well as the experimental conclusions that can be drawn.

The active and reactive power delivered by DGs are determined based on systematic values of voltage and current. The droop controller only generates the reference voltage and frequency for the voltage controller, which produces the reference current for the current controller [44]. Lastly, space vector pulse width modulation (SVPWM) creates controlled signals to deliver the power as either active or reactive to the load within nominal frequency. Further, SVPWM also control the supply voltage to the three-phase voltage source inverter (VSI). Figure 3 illustrates a block diagram of the MG islanding mode.

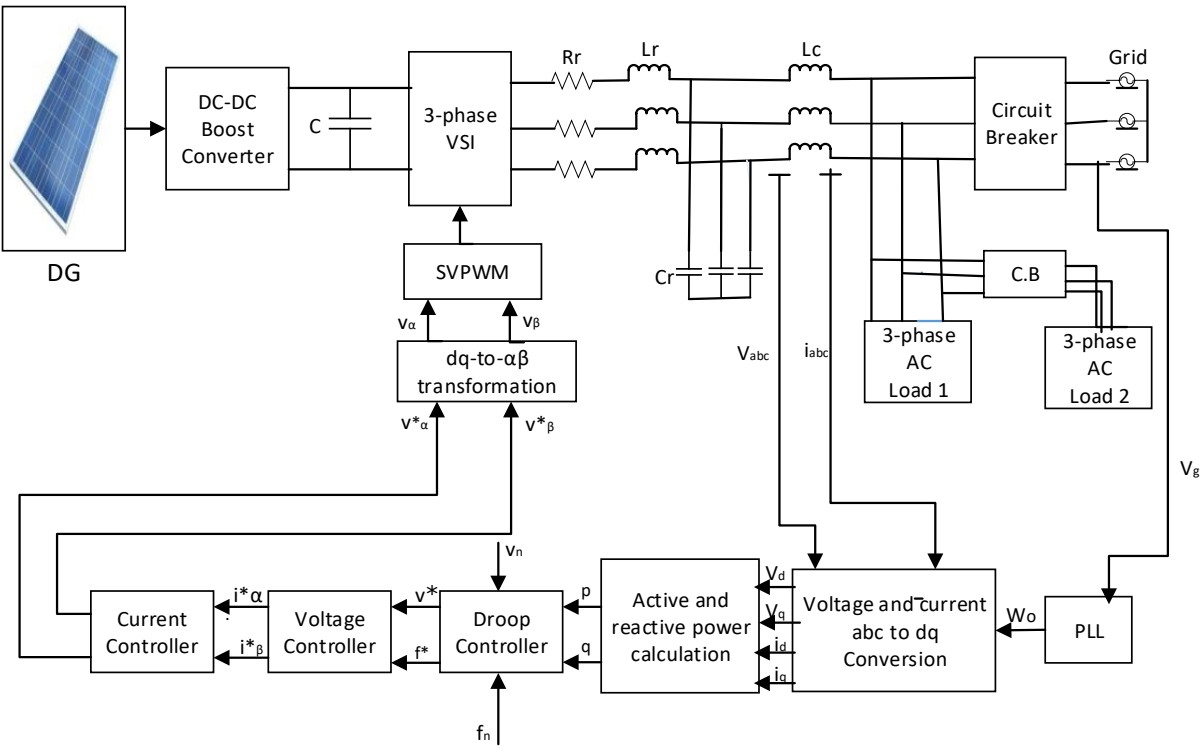

**Figure 3.** Block diagram for islanded MG.

According to Figure 3, $v_d$, $v_q$ are output voltages; $i_d$, $i_q$ are output currents in the direct-quadrature (d-q) form; DC capacitance is denoted by C; $R_f$ is the resistance per-phase; $C_f$ is the capacitance per-phase; and $L_f$ is the inductance per-phase of the output filter. The three-phase MG voltage and current are represented by $v_{abc}$ and $i_{abc}$, respectively; angular frequency is denoted by $\omega_o$ and at the grid output voltage, it is denoted by $V_g$; the active power and reactive power are denoted by $p$ and $q$, respectively; the nominal voltage and nominal frequency are denoted by $v_n$ and $f_n$, respectively; reference voltage and frequency are denoted by $v^*$ and $f^*$, respectively; $i_d^*$, $i_q^*$ are the reference currents and $v_d^*$, $v_q^*$ are the reference voltages in the d-q form; the equivalent voltage signals are denoted by $v_\alpha$ and $v_\beta$ in the $\alpha\beta$ form.

The power circuit dynamics of the MG system under study was mathematically modeled with the help of the Park's transformation, which is determined by the equations of state space as follows [34]:

$$\frac{d}{dt}\begin{bmatrix} i_d \\ i_q \end{bmatrix} = \begin{bmatrix} -\frac{R_f}{L_f} & \omega_o \\ -\omega_o & -\frac{R_f}{L_f} \end{bmatrix}\begin{bmatrix} i_d \\ i_q \end{bmatrix} + \frac{1}{L_f}\left( \begin{bmatrix} v_{vd} & v_d \\ v_{vq} & v_q \end{bmatrix} \right) \tag{15}$$

$$\frac{d}{dt}\begin{bmatrix} v_d \\ v_q \end{bmatrix} = \begin{bmatrix} 0 & \omega_o \\ -\omega_o & 0 \end{bmatrix}\begin{bmatrix} v_{vd} \\ v_{vq} \end{bmatrix} + \frac{1}{C_f}\left( \begin{bmatrix} i_{vd} \\ i_{vq} \end{bmatrix} - \begin{bmatrix} i_d \\ i_q \end{bmatrix} \right) \tag{16}$$

where, $v_a$, $v_b$, $v_c$ and $i_a$, $i_b$, $i_c$ represent the per-phase voltage and current, respectively. $v_{vd}$, $v_{vq}$ represent the output voltage of the filter. $i_{vd}$, $i_{vq}$ represent the output current of the filter.

### 4.1. Voltage and Frequency Control Model

In the islanded mode of an MG, the main grid is not connected to the MG throughout course of its operation. The controlled pulses produced by the control circuit are provided to the VSI. Then, a smooth sinusoidal waveform of voltage is created, which supply the produced energy from DG to load. The PI controllers have two gains: one is proportional gain (Kp) and other is integral gain (Ki). Both gains are tuned by the SRGWO algorithm to improve the dynamic response of the MG system. The block diagram of the strategy of control is shown in Figure 4.

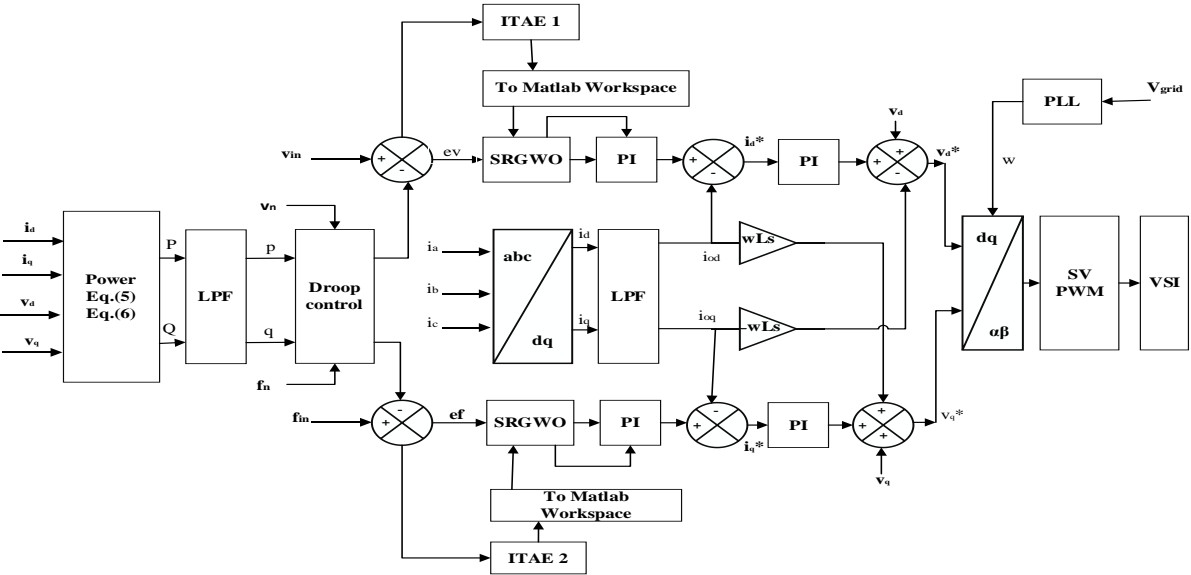

**Figure 4.** Block diagram of control strategy.

In Figure 4, $v_m$ is measured voltage and $f_m$ is measured frequency; the voltage error and frequency error are denoted by $e_v$ and $e_f$, respectively; the nominal voltage and nominal frequency are denoted by $v_n$ and $f_n$, respectively. ITAE is the abbreviation of integral time absolute error, while LPF is the abbreviation of low pass filter.

Initially, the DGs have to generate the voltage and current signals measured in the main grid. Afterwards, these signals are changed to the reference d-q form through Park's transformation using the equation as follows:

$$\begin{bmatrix} v_d \\ v_q \\ v_o \end{bmatrix} = \sqrt{\frac{2}{3}} \begin{bmatrix} \cos\theta & \cos\left(\theta - \frac{2\pi}{3}\right) & \cos\left(\theta + \frac{2\pi}{3}\right) \\ -\sin\theta & -\sin\left(\theta - \frac{2\pi}{3}\right) & -\sin\left(\theta + \frac{2\pi}{3}\right) \\ \frac{1}{\sqrt{2}} & \frac{1}{\sqrt{2}} & \frac{1}{\sqrt{2}} \end{bmatrix} \begin{bmatrix} v_a \\ v_b \\ v_c \end{bmatrix} \tag{17}$$

$$\begin{bmatrix} i_d \\ i_q \\ i_o \end{bmatrix} = \sqrt{\frac{2}{3}} \begin{bmatrix} \cos\theta & \cos\left(\theta - \frac{2\pi}{3}\right) & \cos\left(\theta + \frac{2\pi}{3}\right) \\ -\sin\theta & -\sin\left(\theta - \frac{2\pi}{3}\right) & -\sin\left(\theta + \frac{2\pi}{3}\right) \\ \frac{1}{\sqrt{2}} & \frac{1}{\sqrt{2}} & \frac{1}{\sqrt{2}} \end{bmatrix} \begin{bmatrix} i_a \\ i_b \\ i_c \end{bmatrix} \tag{18}$$

The generated active power and reactive power of DGs is already converted to d-q form. Hence, the consumed power in the form of d-q, is determined as follows [24]:

$$P = v_d i_d + v_d i_q \tag{19}$$

$$Q = v_d i_q - v_q i_q \tag{20}$$

where the active power is denoted by *P*, and the reactive power is denoted by *Q* before LPF.

The LPF is utilized in this paper as given in Equations (21) and (22). This would help to obtain the basic elements of *p* (the active power), *q* (the reactive power), and decrease effect of current as well as the power regulator.

$$p = \frac{\omega_c}{S + \omega_c} P \tag{21}$$

$$q = \frac{\omega_c}{S + \omega_c} Q \tag{22}$$

where the cut-off frequency of the filter is denoted by $\omega_c$, and Laplace transform operator is denoted by *S*.

### 4.2. Voltage and Frequency Controller

The droop controller created the reference voltage and reference frequency for the voltage controller in this study. The purpose of the regulator was to obtain the required frequency and voltage values by eliminating the error due to load changes or DG insertion. Two PI controllers were used in this controller, whose four gains were tuned using an intelligent meta-heuristic method called the SRGWO algorithm. Mathematically, the dynamic of the PI controller can be shown as follows:

$$i_d^* = (v^* - v_n)\left(k_{pv} + \frac{k_{iv}}{s}\right) \tag{23}$$

$$i_q^* = (f^* - f_n)\left(k_{pf} + \frac{k_{if}}{s}\right) \tag{24}$$

It produced a signal of mention current ($i_d^*$ and $i_q^*$) for the current controller. The output mention currents ($i_d^*$ and $i_q^*$) could be controlled with the help of reducing the signal of voltage error ($e_v$), according to control loop of voltage, as in (23) and (24). Therefore, both power flows were optimized by the DG converter by controlling the output mention currents of the voltage control loop.

### 4.3. Current Controller

The current controller used traditional PI controllers to monitor the output of PWM at mention points $i_d^*$ and $i_q^*$. In order to increase the stability of the current controller based on PI, the decoupling operation was accepted by means of current feed-forward reimbursement. It was attained through regarding inverter set currents ($i_d^*$, $i_q^*$) in its place of output measured currents ($i_d$, $i_q$) [45]. In addition, according to the control composition illustrated in Figure 4, the output voltage signal equation of current loop is written as follows:

$$v_d^* = i_d^* - i_{vd}\left(k_{pv} + \frac{k_{iv}}{s}\right) - \omega.L_f.i_{vq} + v_d \tag{25}$$

$$v_q^* = i_q^* - i_{vd}\left(k_{pf} + \frac{k_{if}}{s}\right) + \omega.L_f.i_{vq} + v_q \tag{26}$$

Since PI tuning based on the SRGWO algorithm was used to reduce error in the voltage regulator, it was not necessary to optimize the parameters for the current regulator. Therefore, two fixed gains for the PI regulators were used to minimize the current error. Space pulse width modulation (SVPWM) is received from the output of the current regulator in the αβ form of the mention. It generates the controlled pulses that are used to disable the VSI so as to supply a power controller to the load with the best power quality. A flowchart of the proposed algorithm execution in the MG controller is illustrated in Figure 5.

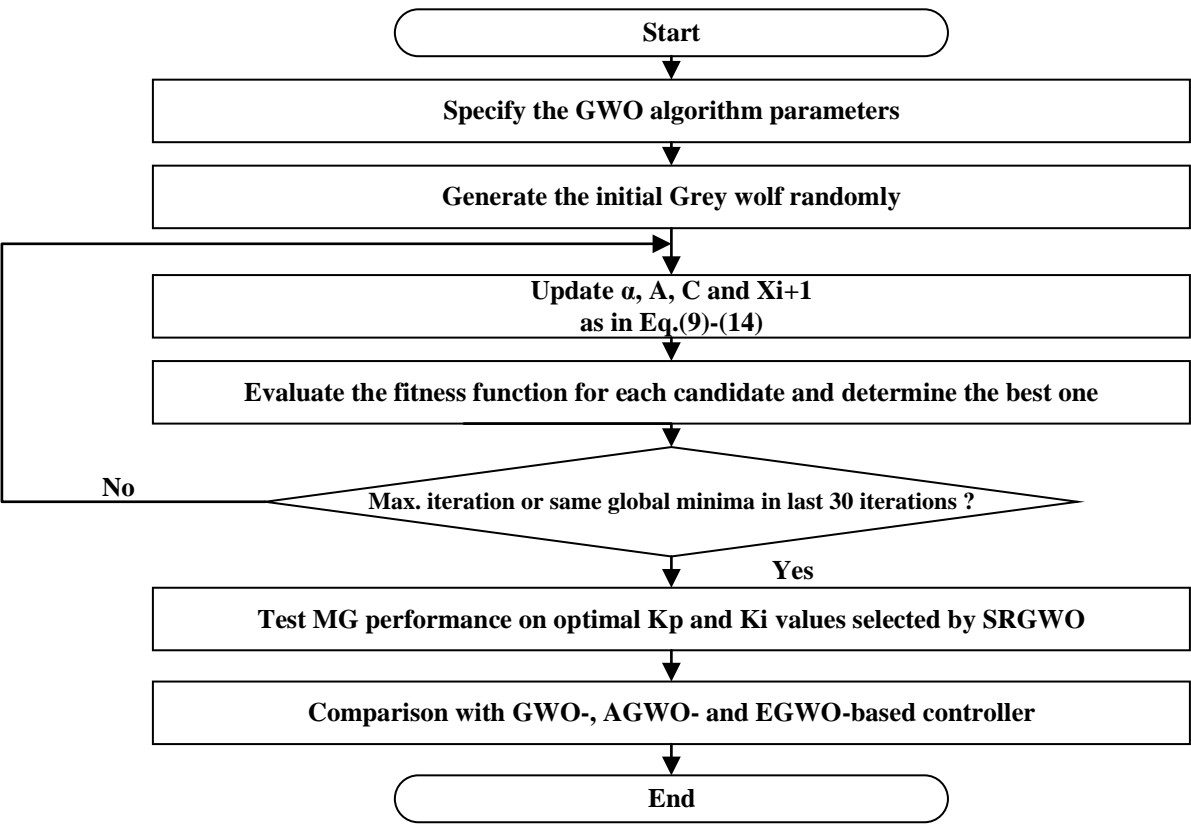

**Figure 5.** Flowchart of the proposed algorithm.

## 5. Problem Formation

In this study, the PI gains controller was employed in the control loop of voltage using a newly introduced meta-heuristics algorithm named SRGWO, so as to resolve the given problems. The study accomplished the lowest value of FF with the help of the proposed algorithm. The most common mechanism for minimizing control purposes in research survey is ITAE. This is due to fact that ITAE enables smoother execution and better results than its competitors, e.g., integrated absolute error (IAE), integrated square error (ISE), and integrated time square error (ITSE) [39,40]. Therefore, ITAE is considered to have the best fitness function (FF) for optimizing voltage and frequency response.

Mathematically, ITAE is determined as:

$$ITAE = \int_0^\infty t|e|dt \tag{27}$$

where time is denoted by t, the error is denoted by e(t), and the minimum is denoted by Min, which is the variation between the controlled variable and the mention value. The FF is considered to be an easy mathematic addition to the voltage and frequency integration functions of ITAE 1 and ITAE 2, and is calculated as follows:

$$FF = Min\left\{ \int_0^\infty t \times |e_v|dt + \int_0^\infty t \times |e_f|dt \right\} \tag{28}$$

The low FF value found in this study guarantees that the best selection of PI gains was found. This further guarantees the best dynamic achievement of the tested islanded MG system. The values of ITAE for the two PI controllers were calculated and fed into the MATLAB workspace. Afterwards, the optimal values of PI parameters are obtained for the proposed algorithm, variants of GWO, PSO and GSA. Later, the optimal values of the PI parameters were then allocated to the controllers in the model (MATLAB/SIMULINK).

Therefore, the proposed controller is optimally dynamic (receiving optimal values whenever a change in system's parameter occurs) during the complete operational response of the investigated MG system.

## 6. Results and Discussions

The SRGWO algorithm was used in this paper to select the best values for the PI coefficient gain of the MG in an islanded mode system through minimizing specified objective function. The outcomes were also later compared with controllers based on GWO, AGWO, and EGWO for the same working situations. For comparison, the particle counts and 50 iterations were defined for each algorithm.

### 6.1. Frequency and Voltage Control at the time of DG Insertion and Load Change

Due to frequency dives and reduced voltage, to achieve the rated frequency and voltage of the investigated MG network during DG insertion and load changes, a controller based on the SRGWO algorithm must select the better gain value for PI controllers. This was obtained using four various intelligence meta-heuristic algorithms (SRGWO, GWO, AGWO, and EGWO) to minimize FF. The main objective was to minimize FF, and therefore, the minimum values were considered to be best value. The convergence behavior of the four investigated methods under the same working situations. is illustrated in Figure 6.

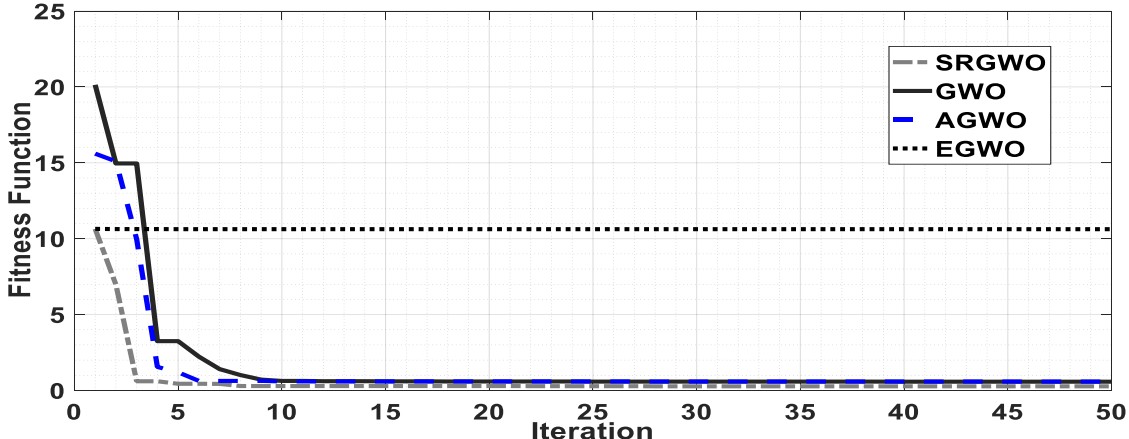

**Figure 6.** The convergence curves.

In Figure 6, the AGWO provides a much better optimum solution than GWO, but with less flexibility than GWO. Meanwhile, it has been established that controllers based on the SRGWO algorithm achieve faster and better optimal solutions than controllers based on the GWO, AGWO, and EGWO algorithms. Table 5 shows the optimized final value and iteration numbers where the minimum FF was achieved.

**Table 5.** The convergence values of SRGWO, GWO, AGWO, and EGWO.

| Controller Type | FF Minimum Value | Obtained Iteration Which Is Minimum Value |
|---|---|---|
| SRGWO | 0.2724 | 28 |
| GWO | 0.5778 | 40 |
| AGWO | 0.5885 | 44 |
| EGWO | 10.6265 | 41 |

When the simulation was performed, this search process ended once the optimal value of objective function was obtained, or after the indicated number of iterations had been executed. In the proposed algorithm, the value of maximum iterations has been set to 50.

The optimized parameters achieved the final value, i.e., the two gains are $K_{pv}$ and $K_{iv}$ for the PI voltage controller, and $K_{pf}$ and $K_{if}$ for the PI frequency controller, for the controllers based on each algorithm in this study. These figures are presented in Table 6.

**Table 6.** The parameters of optimized PI controller.

| Optimization | $K_{pv}$ | $K_{iv}$ | $K_{pf}$ | $K_{if}$ |
|---|---|---|---|---|
| SRGWO | $8.93 \times 10^{-2}$ | $1.56 \times 10^1$ | $-8.70 \times 10^{-3}$ | $-8.97 \times 10^{-4}$ |
| GWO | $-1.43 \times 10^{-1}$ | $-1.0 \times 10^1$ | $-1.91$ | $3.44 \times 10^1$ |
| AGWO | $-1.31 \times 10^{-1}$ | $-1.0 \times 10^{-1}$ | $-1.18$ | $3.70 \times 10^1$ |
| EGWO | $3.15 \times 10^1$ | $2.20 \times 10^1$ | $4.14 \times 10^1$ | $6.19 \times 10^1$ |

In addition, at the beginning of the simulation, an evaluation of the performance of the control system was implemented, when a load of 50 kW (20 kVAR) was connected with DG. The DG unit employed a voltage-frequency power control model dependent on the SRGWO algorithm to maintain voltage and prevent serious frequency variation due to the rapid insertion of the DG. After that, another 30 kW (20 kVAR) load was injected into system in 0.4 s.

Figure 7a,b shows, under the same working circumstances, the comparison of active power (kW) and reactive power (kVAR) variations among the four various regulators.

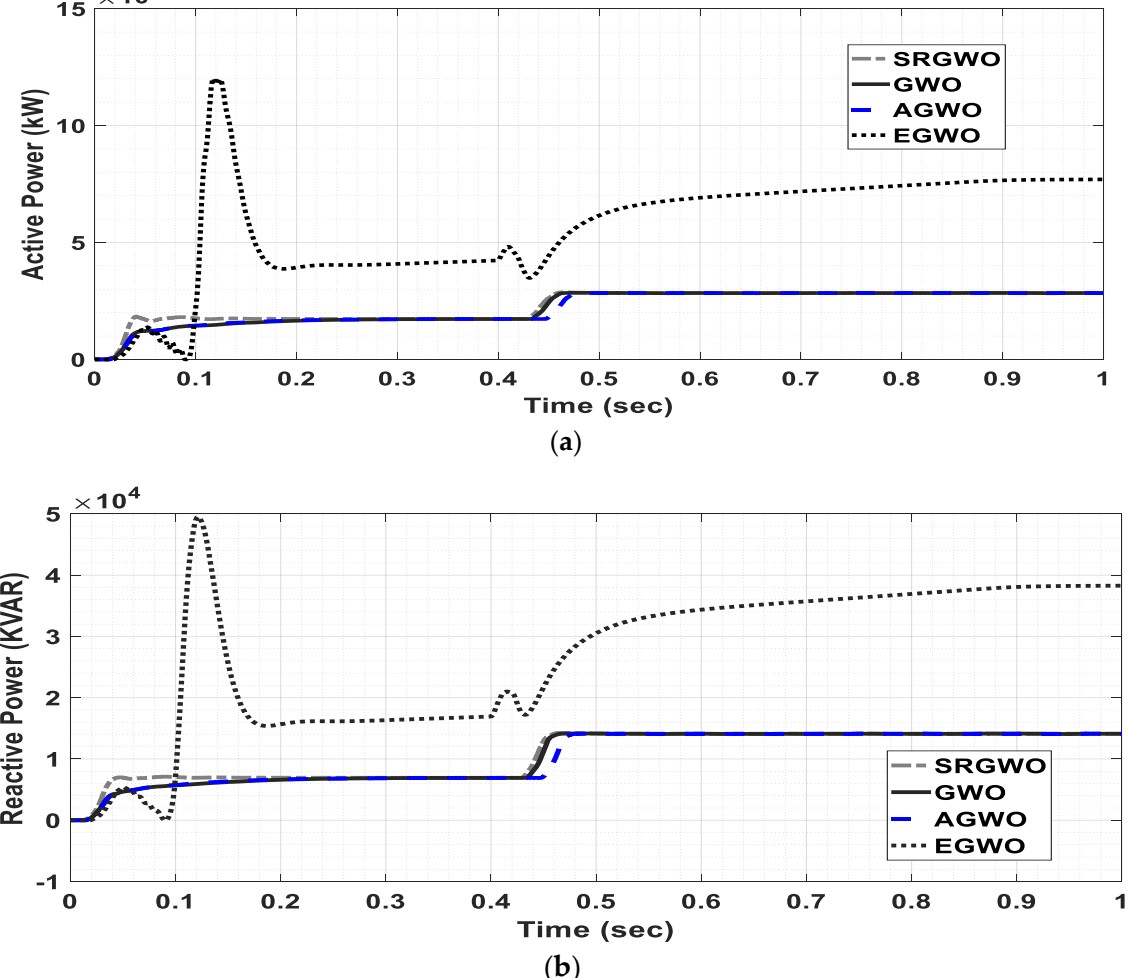

**Figure 7.** (**a**,**b**) The comparison of SRGWO with AGWO-, EGWO-, and GWO-based controllers (**a**) Active Power (**b**) Reactive Power.

It may be noted that when another load of 30 kW, 20 kVAR was injected in 0.4 s into the system, the voltage reduced quickly. This has happened because of more voltage drop across the filter components, and more load resistance. The controller was running to instantly recover the rated voltage and frequency of the system, as shown in Figure 8a,b.

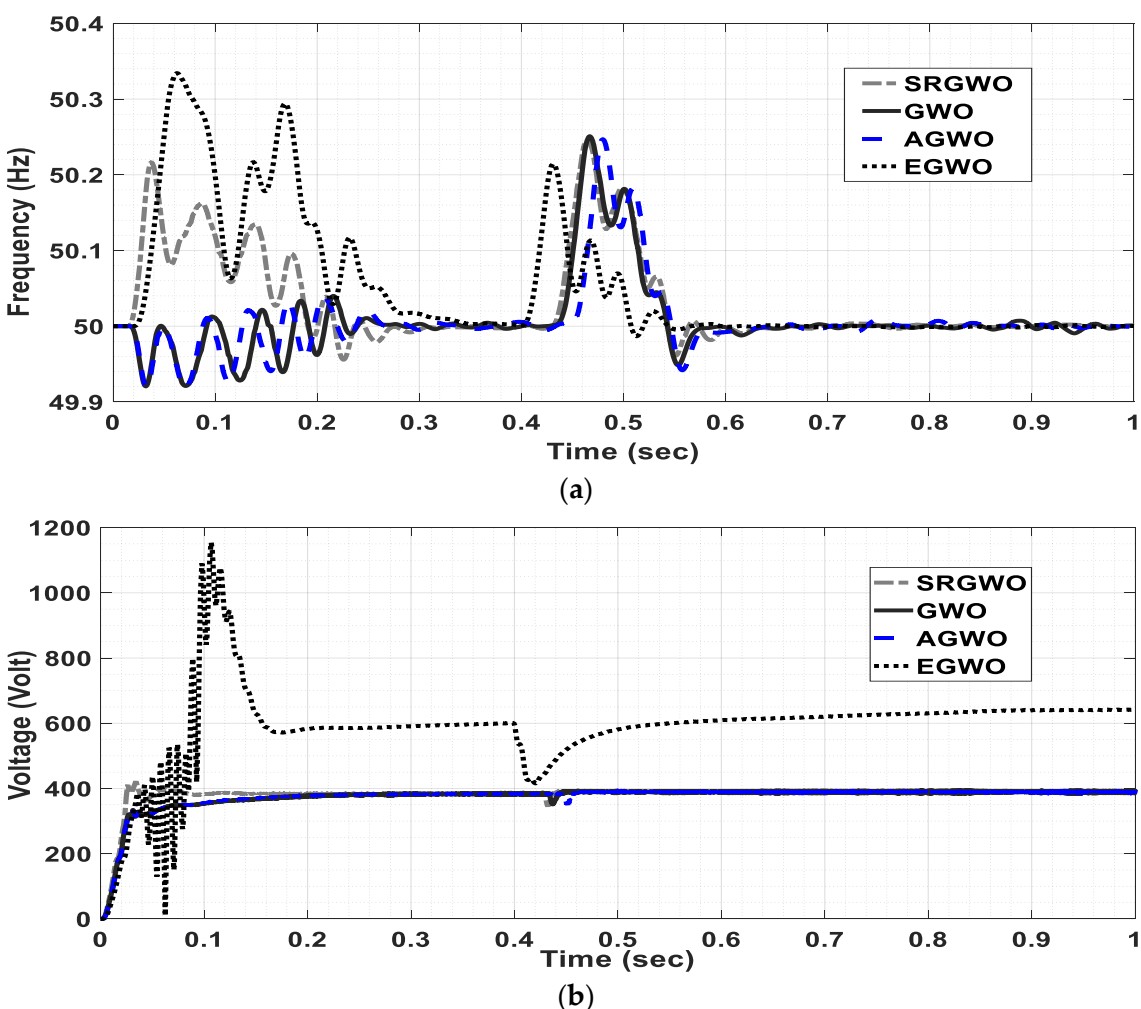

**Figure 8.** (**a**,**b**) The comparison of SRGWO with EGWO, AGWO and GWO (**a**) Frequency regulation (**b**) Voltage regulation.

It is quite tough, but possible, to achieve online optimization in real-time, in practical applications and under various working situations, due to the process of long search, random load switching, and simulation time. Consequently, the parameters optimized for operating conditions were selected using SRGWO, GWO, AGWO, and EGWO, i.e., load changes and DG insertion during the simulation worked. The main advantages of this technique include the smooth adjustment of optimized parameters. Furthermore, the achieved parameters were good in all working situations. As the MATLAB editor ran optimization algorithm code, the application algorithm began to search for optimal parameters of PI by reducing the cost function.

### 6.2. Steady-State and Dynamic Response

One of the key parameters that must be eliminated during the islanded mode of operation is the sinusoidal nature of the output voltage and current of the inverter. The optimal parameters of the PI controller provide a smooth, and sinusoidal waveform, which minimizes the total harmonic distortion (THD). The controller based on the SRGWO

algorithm given a very small percentage value of THD by a fast Fourier transform analysis of the waveform of output current. which demonstrates its performance in controlling the frequency and voltage of the investigated MG system, along with the high quality of its power. In Table 6, the algorithms compare the proposed SRGWO-based controllers with GWO-, AGWO-, and EGWO-based controllers. It shows the current waveform THD level of the output of 15 cycles on 50 Hz of fundamental frequency. According to IEEE standard 1547–2003 [46], the percentage of THD allowed in an electrical supplier must be always less than 5%. The THD in percentage achieved by controllers based on the proposed algorithm, GWO, AGWO, and EGWO, due to the insert of DG and load changes, are shown in Table 7.

**Table 7.** The output current THD for GWO, AGWO, EGWO and SRGWO algorithm-based controllers.

| Algorithm-Based Controller Type | Percentage of THD DG Insertion | Percentage of THD Load Changes |
|---|---|---|
| GWO | 1.29 | 0.13 |
| AGWO | 1.3 | 0.16 |
| EGWO | 13.47 | 1.39 |
| SRGWO | 0.35 | 0.07 |

## 7. Conclusions

A novel optimal voltage and frequency control for DG units in a microgrid is proposed as a simple and an efficient algorithm. Twenty-three benchmark functions have been tested using the SRGWO method. The results have proved that the implementation of the proposed algorithm showed better performance in real power systems as compared to other GWO, PSO, AGWO, EGWO, and GSA algorithms. To enhance the capability of the proposed algorithm, the PI controller parameters were optimized by minimizing the error, which increased its efficacy. Additionally, the proposed algorithm-based control architecture provides a superior and optimal response compared to traditional GWO-, adaptive GWO- and enhanced GWO-based controllers, which makes it applicable to modern power systems, with high-speed response to disturbances. It also achieves a better minimum final optimized value of the FF as compared to GWO, AGWO, and EGWO, which ensures a high-quality solution for the stated optimization problem. A power quality analysis established that the SRGWO-based controller provides the least THD (%) as compared to GWO-, AGWO-, and EGWO-based controllers, and thus meets the IEEE standard 1547–2003.

The voltage and frequency magnitudes obtained by SRGWO offer minimum overshoot and settling time for the DG insertion condition, as well as the load variation condition. This leads to high-speed recovery of stability in a real-time power system. Furthermore, the proposed method has achieved an optimal fitness function and better power quality, which will ensure the economic and secure operation of power systems. In the future, real-time voltage and frequency control using machine learning will be performed using systems with a high penetration of renewable energy. The implementation of the proposed algorithm to a more complex power system network, such as hybrid renewable energy resources, as well as a real-time power networks, such as experimental prototypes, is suggested for future work.

**Author Contributions:** Conceptualization, A.A.A., A.N., E.M. and F.U.; investigation F.U., A.N., A.A.S. and R.u.H.; formal analysis, A.A.A., F.U., R.u.H. and E.M.; writing—original draft preparation, A.A.A.; visualization, A.N. and A.A.S.; writing—review and editing, X.H.; supervision, X.H. All authors have read and agreed to the published version of the manuscript.

**Funding:** This research received no external funding.

**Data Availability Statement:** Not applicable.

**Conflicts of Interest:** The authors declare no conflict of interest.

**List of Symbols and Abbreviations:**

| | |
|---|---|
| MG | Microgrid |
| PI | Proportional Integral |
| GWO | Gray Wolf Optimization |
| SRGWO | Square Root Gray Wolf Optimization |
| EGWO | Enhanced Gray Wolf Optimization |
| AGWO | Augmented Gray Wolf Optimization |
| mGWO | Modified Gray Wolf Optimization |
| RES | Renewable Energy Source |
| PV | Photovoltaic |
| VSI | Voltage Source Inverter |
| Z-N | Ziegler-Nichols |
| AI | Artificial Intelligence |
| FL | Fuzzy Logic |
| GA | Genetic Algorithm |
| DG | Distributed Generator |
| PSO | Particle Swarm Optimization |
| SVPWM | Space Vector Pulse Width Modulation |
| PWM | Pulse Width Modulation |
| ITAE | Integral Time Absolute Error |
| IAE | Integrated Absolute Error |
| ISE | Integrated Square Error |
| ITSE | Integrated Time Square Error |
| FF | Fitness Function |
| LPF | Low Pass Filter |
| PLL | Phase Locked Loop |
| THD | Total Harmonic Distortion |

**Symbols and Parameters**

| | |
|---|---|
| $v_d$ and $v_q$ | output voltages in d-q form |
| $i_d$ and $i_q$ | output currents in d-q form |
| C | DC capacitance |
| $R_f$ | Filter resistance per-phase |
| $C_f$ | Filter capacitance per-phase |
| $L_f$ | Filter inductance per-phase |
| $v_{abc}$ | 3-phase MG voltage |
| $i_{abc}$ | 3-phase MG current |
| $\omega_o$ | Angular frequency |
| $V_g$ | Grid output voltage |
| $v_n$ | Nominal voltage |
| $f_n$ | Nominal frequency |
| $v^*$ | Reference voltage |
| $f^*$ | Reference frequency |
| $i_d^*$ and $i_q^*$ | Reference current in d-q form |
| $v_d^*$ and $v_q^*$ | Reference voltage in d-q form |
| $v_\alpha$ and $v_\beta$ | Voltage in the $\alpha\beta$ form |
| $v_a, v_b, v_c$ | Per-phase voltage |
| $i_a, i_b, i_{(c)}$ | Per-phase current |
| Kp | Proportional gain |
| Ki | Integral gain |
| $v_m$ | Measured voltage |
| $f_m$ | Measured frequency |
| $e_{(v)}$ | Error voltage |

| | |
|---|---|
| e_(f) | Error frequency |
| (X_i) | Position vector of gray wolf |
| (X_pi) | Position vector of quarry |
| A and C | Coefficient Vectors |
| r_1 and r_2 | Uniform randomly distributed vectors |
| S | Operator of Laplace transform |
| Kpv and Kiv | Gain parameters of the PI voltage controller |

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
