# Peer review of "Optimal Solution for Frequency and Voltage Control of an Islanded Microgrid Using Square Root Gray Wolf Optimization"

_electronics, doi:10.3390/electronics11223644_

Round 1

Reviewer 1 Report

There are serious problems associated with the current version as follows:

1- The context needs to be improved since there are some mistakes linguistically and grammatically.

2- A more detailed abstract with more quantitative data is needed. Clearly mention the aim and main findings of your study. In the abstract, provide some info on the importance of the study subject before giving the study content.

3- The literature review is not completed, and many important works have been ignored. for example, this work that has surveys optimization methods: https://doi.org/10.1016/j.seta.2021.101370

4- The result must be compared with the top-tier works in the literature. Also, what is the superiority of the presented method over other techniques, including learning-based methods.

5-The quality of the figures could be improved.

Reviewer 2 Report

The current paper deals with the frequency and voltage control issues in microgrids. the topic is interesting. The efforts in the paper are appreciated.  The following comments are necessary to improve the paper quality and presentation: 

1. Update both the abstract and conclusion section with the paper contribution.

2. The research gaps need more highlights and therefore the paper contributes to solving these gaps. 

4. Editing attention is needed. for example Line 151 ( A gravitation search... etc ) the sentence is presented without a verb. 

5. Avoid multiple citations. only mention critical issues for each reference  

6. Why is GWO among the exit optimizers?

7. Add numerical statistical evaluation of the proposed modified version of GWO for the target problem.

8. Improve the quality of the figure as  Figure 5. 

9. Discussion of the numerical simulation is poor. 

10. Application on PI controllers is very limited. Extend to PID controller.  11. Discuss the power quality issues for various disturbances in more detail. 

12. Add the future trend of this study on both the problem side and solution methodology side. 

Reviewer 3 Report

The paper presents the optimal solution to voltage control  using a heuristic approach. In the introductory part of the paper the authors discuss power generation problem and the other inherently connected one, namely the power loss case, resulting in giving rise to microgrids. The most severe one is frequency instability, and loss of synchronization between the microgrid and the remaining power system. 

To support the improvement of the performance of such a microgrid , the authors refer to standard control techniques, as PI, discussing the need for tuning parameters of controllers, giving appropriate references from the background. 

The genetic-based approaches are disregarded as they admit to larger resource requirement and their time-consuming calculations, eventually leading to meta-heuristics. 

The discussion presented there is complete and nicely set. Contributions are clearly stated. 

However, please add also a novelty statement. 

The notation in tables 1, 2, etc (BTW - Table 2 is used twice) should be given in 10^{} instead in E. 

I believe section 3.1 is surplus, as the comparison between the set of methods can be easily cited from the literature to preserve readibility of the paper. 

What the arguments of the optimization (decision variables) in (28) are? They should be listed below MIN. 

Figure 5 is not intellegible. 

No stopping criteria are discussed and their impact on the overall performance is not given. 

The presentation of the results is well-supported and vast simulation campaign is given. The comparison between a family of GWO approaches is found to be sufficient. 

Round 2

Reviewer 1 Report

Hello,

I went through your answers to all reviewers' comments. Unfortunately, 

most of the material is duplicated by simply copying and pasting for the other reviewer. Hence, I still want you to carefully read and address my comments. MAKE SURE that you exactly answer the comments.

Reviewer 2 Report

No further comments 

Author Response

The authors thank the reviewer for his kind review and for showing satisfaction with the manuscript's quality.

Reviewer 3 Report

Thank you for providing the response to my queries. As far as the response itself is concerned - there is nothing as more or less optimal. Just optimal in some sense. 

I have no further comments on the paper. Thank you for revising the paper. Good luck with the review process

Author Response

The authors appreciated and thanked the reviewer as he spared time to review this article and provided his guidance in improving the manuscript.

Round 3

Reviewer 1 Report

I have no further comments.

Thanks.